# Genomic Correlates of DNA Damage in Breast Cancer Subtypes

**DOI:** 10.3390/cancers13092117

**Published:** 2021-04-27

**Authors:** Esther Cabañas Morafraile, Javier Pérez-Peña, Jesús Fuentes-Antrás, Aránzazu Manzano, Pedro Pérez-Segura, Atanasio Pandiella, Eva M. Galán-Moya, Alberto Ocaña

**Affiliations:** 1Experimental Therapeutics Unit, Hospital Clínico San Carlos (HCSC), Instituto de Investigación Sanitaria San Carlos (IdISSC) and Centro de Investigación Biomédica en Red en Oncología (CIBERONC), 28040 Madrid, Spain; esther.cabanas@salud.madrid.org (E.C.M.); jfuentesa@salud.madrid.org (J.F.-A.); aranzazu.manzano@salud.madrid.org (A.M.); pedro.perez@salud.madrid.org (P.P.-S.); 2Instituto de Biología Molecular y Celular del Cáncer del CSIC, IBSAL and CIBERONC, 37007 Salamanca, Spain; jppena@sescam.jccm.es (J.P.-P.); atanasio@usal.es (A.P.); 3Translational Oncology Laboratory, Centro Regional de Investigaciones Biomédicas (CRIB) and Nursery School, Campus de Albacete, Universidad de Castilla-La Mancha, 02008 Albacete, Spain; evamaria.galan@uclm.es

**Keywords:** breast cancer, DNA damage response (DDR), *ATR*, *Fanconi Anemia*, *ATM*, *BARD1*, biomarkers, genomic signatures

## Abstract

**Simple Summary:**

Breast cancer (BC) is the most common invasive tumor in women and the second leading cause of cancer-related death. Therefore, identification of druggable targets to improve current therapies and overcome resistance is a major goal. In this work, we performed an in silico analysis of transcriptomic datasets in breast cancer, and focused on those involved in DNA damage, as were clearly upregulated using gene set enrichment analyses (GSEA), particular the following pathways: *ATM/ATR, BARD1* and *Fanconi Anemia*. *BHLHE40, RFWD2, BRIP1, PRKDC, NBN, RNF8, FANCD2, RAD1, BLM, DCLRE1C, UBE2T, CSTF1, MCM7, RFC4, YWHAB, YWHAZ, CDC6, CCNE1,* and *FANCI* genes were amplified/overexpressed in BC, and correlated with detrimental prognosis. Finally, we selected the best transcriptomic signature of genes within this function that associated with clinical outcome to identify functional genomic correlates of outcome.

**Abstract:**

Among the described druggable vulnerabilities, acting on the DNA repair mechanism has gained momentum, with the approval of PARP inhibitors in several indications, including breast cancer. However, beyond the mere presence of *BRCA1/BRCA2* mutations, the identification of additional biomarkers that would help to select tumors with an extreme dependence on DNA repair machinery would help to stratify therapeutic decisions. Gene set enrichment analyses (GSEA) using public datasets evaluating expression values between normal breast tissue and breast cancer identified a set of upregulated genes. Genes included in different pathways, such as *ATM/ATR, BARD1,* and *Fanconi Anemia*, which are involved in the DNA damage response, were selected and confirmed using molecular alterations data contained at cBioportal. Nineteen genes from these gene sets were identified to be amplified and upregulated in breast cancer but only five of them *NBN*, *PRKDC, RFWD2, UBE2T,* and *YWHAZ* meet criteria in all breast cancer molecular subtypes. Correlation of the selected genes with prognosis (relapse free survival, RFS, and overall survival, OS) was performed using the KM Plotter Online Tool. In last place, we selected the best signature of genes within this process whose upregulation can be indicative of a more aggressive phenotype and linked with worse outcome. In summary, we identify genomic correlates within DNA damage pathway associated with prognosis in breast cancer.

## 1. Introduction

Cancer is characterized by a wide range of genomic alterations, with some of them involved in the oncogenic process [1]. The heterogeneous nature of the disease is linked with resistance to different therapeutics, as not all tumor cells express targetable vulnerabilities, and some develop mechanisms to escape therapeutic induction of cell death [1,2,3]. Among the different deregulated functions described as involved in cancer, regulation of DNA repair mechanisms has been reported as one with great potential for pharmacological intervention [4,5]. Genes involved in the process of DNA repair have been described as functionally dysregulated in several tumors, contributing substantially to the high grade of genomic instability observed in some cancers [6,7]. Germline mutations at BRCA1 and BRCA2 have been described in several solid tumors, such as breast, prostate or pancreatic cancer; and agents acting on the protein PARP have shown activity in this specific population [6,7,8,9]. Tumors that lack functional BRCA proteins have impaired the homologous recombination (HR) pathway and use alternative mechanisms for DNA repair, such as the non-homologous end joining (NHEJ) pathway [6]. Targeting components of the NHEJ (e.g., the protein PARP) induce a synthetic lethality approach in cells with germline inactivating mutations at BRCA1 and BRCA2 [6]. 

Breast cancer is a heterogeneous disease, not only by its transcriptomic profile with different breast cancer subtypes already described, but at a genomic level, where different modifications, including mutations or amplifications at relevant genes, can classify tumors with different clinical outcome, and potential for therapeutic intervention [1,10]. Beyond the classical breast cancer subtypes based on transcriptomic data, some of these genomic alterations are expressed in different subgroups in an agnostic manner, such as fusions at NTRK gene or BRCA1/2 mutations, among others [11]. On the other hand, some subtypes share some phenotypic characteristics. For instance, those tumors that lack the expression of the estrogen and progesterone receptors (ER and PR respectively) and the HER2 membrane receptor, and are therefore termed triple negative, have more genomic instability, respond better to platinum agents, and harbor an augmented presence of activated immune cells [11,12]. 

Targeting DNA damage response (DDR) has shown efficacy in tumors where these pathways are dysregulated, and this can be recognized by the presence of mutations in some genes [13,14]. In addition to the presence of germline mutations at BRCA1 and BRCA2, recent studies have shown that mutations at somatic BRCA1, BRCA2, or germline PALB2, can also predict response to the PARP inhibitor olaparib in breast cancer [15], opening the door for the evaluation of other genes of the homologous recombinant pathway as potential biomarkers. In addition, these tumors also respond better to agents that induce DNA damage like platinum compounds, observation already included in therapeutic guidelines [16]. However, beyond the presence of the mutations mentioned before, the detection of additional biomarkers that would aid in the recognition of tumors with an extreme dependence on DNA damage would help to select and stratify therapeutic decisions. 

The use of genomic biomarkers to optimize treatment based on a risk assessment of relapse has reached the clinical setting, as is the case with the use of genomic panels to stratify chemotherapy in early stage breast cancer [17]. A similar approach, but based on a genomic vulnerability and a potential family of targeted agents against that vulnerability, has not been exploited yet, but undoubtedly would improve patient care. For the time being, new compounds have only been developed based on the single expression of the target without analyzing the global biological dysfunction to which that target is acting. 

In our article, we aimed to explore genomic correlates that were associated with DNA damage response in all breast cancer subtypes. Through the evaluation of different genomic datasets and data mining, we identified a transcriptomic signature that selected patients with a particular detrimental outcome. Of note, several of the reported genes were amplified in breast cancer and could therefore be easily analyzed. Finally, the selection of specific signatures was able to differentially discriminate tumors with poor prognosis.

## 2. Materials and Methods

### 2.1. Whole Genome Transcription Profiling and Gene Set Enrichment Analyses

mRNA level data from normal breast tissues and basal-like tissues were extracted from a public dataset (GEO DataSet accession numbers: GSE21422, GSE26910, GSE3744, GSE65194, and GSE42568). Affymetrix CEL files were downloaded and analyzed with Affymetrix Expression Console. We further performed gene set enrichment analysis (GSEA) to identify transcription related gene sets that varied between normal and basal-like tissues (Date of analysis: May of 2018). 195 Gene sets were collected from Pathway Interaction Database (PID) [18], via the NDEx database (www.ndexbio.org) [19]; the data were analyzed by GSEA with parameter set to 1.000 gene-set permutations. The enrichment score corresponds to a weighted Kolmogorov–Smirnov-like statistic and reflects the extent to which the gene set is overrepresented at the extreme (i.e. top or bottom) of the entire ranked list. If the enrichment score is positive (e.g., the gene set is overrepresented by top ranked genes), then the gene set is considered upregulated while it is considered downregulated if the score is negative. The networks were construct by Cytoscape software (version 3.4.0) [20]. Affymetrix CHP files were analyzed with Affymetrix Transcriptome Analysis Console 3. Only genes with a maximum of 0.05 *p*-value differential expression between the control and tumor were selected.

### 2.2. Evaluation of Molecular Alterations

We used data contained at cBioportal (www.cbioportal.org) (accessed in October 2020) [21,22], Breast Invasive Carcinoma TCGA (*n* = 816) to explore the role of amplifications and mutations in the identified genes.

### 2.3. Construction and Analysis of PPI Networks and Functional Annotation

We used the online tool STRING (http://www.string-db.org, accessed in October 2020) [23] to construct interactome maps of both amplificated and overexpressed genes in all subtypes of breast cancer (STRING v10 data accessed: September 2020). The indicated network properties include: nodes: number of proteins in the network; edges: number of interactions; node degree: average number of interactions; clustering coefficient: indicates the tendency of the network to form clusters. The closer the local clustering coefficient is to 1, the more likely it is for the network to form clusters; PPI enrichment *p* value: indicates the statistical significance. Proteins are considered hubs when they have more interactions than the average (nº interactions > node degree).

### 2.4. Expression Analyses

The analysis comparing the expression level of genes between normal breast samples (*n* = 291) and breast invasive carcinoma samples (*n* = 1085) including the luminal A (*n* = 415), luminal B (*n* = 194), HER2+ ( *n* = 66), and basal-like (*n* = 135) subgroups, was performed using GTEx and TCGA data in GEPIA2 [24].

### 2.5. Outcome Analyses

The KM Plotter Online Tool (http://www.kmplot.com, accessed in October 2020) [25] was used to evaluate the relationship between the expression of different genes and patient clinical outcomes in different breast cancer subtypes. This open access database allows us to investigate overall survival (OS) and relapse-free survival (RFS) in basal-like, luminal A, luminal B, HER2+, and basal-like breast cancers. Breast cancer subtypes were defined as follow: basal-like as ESR1-/HER2-, luminal A as ESR1+/HER2-/MKI67 low, luminal B as ESR1+/HER2-/MKI67 high, and ESR1+/HER2+, HER2+ as ESR1-/HER2+, and finally triple negative as ER-/PR-/HER2-.

### 2.6. Data Availability

The datasets, accessed in May 2018 and analyzed in the current study, are available in the Gene Expression Omnibus (https://www.ncbi.nlm.nih.gov/geo/query/acc.cgi?acc=, accessed in October 2020) webpage with the GEO accession number: GSE21422, GSE26910, GSE3744, GSE65194, and GSE42568. The breast cancer transcriptome profile generated is available by request.

## 3. Results

### 3.1. Transcriptomic Mapping and Gene Set Enrichment Analyses Identify DNA Repair Pathways as Upregulated in Breast Cancer

To explore gene expression alterations in breast cancer, we performed a gene set enrichment analysis comparing normal breast tissue with breast cancer using publicly available datasets (GSE21422, GSE26910, GSE3744, GSE65194, and GSE42568). As can be seen in Figure 1A, the functional transcriptomic map generated from this comparison showed all gene sets represented with circles. In this picture, the circle size correlates with the number of genes, and the color intensity with the functional enrichment by its statistical significance. Among them, we focused on those related with DDR pathways, including those classified by gene ontology as *ATR, Fanconi Anemia, ATM*, and *BARD1* pathways, where a strong enrichment score (ES) and a low false discovery rate (FDR) was observed (*ATR* pathway; NES: 1,93; FDR: 0.002; *Fanconi Anemia* pathway; NES: 1.869; FDR: 0.004; *BARD1* pathway; NES: 1.78; FDR: 0.013 *ATM* pathway; NES: 1.746; FDR: 0.015 ) (Figure 1B). Then, the expression of these four DNA damage gene sets was analyzed in the four described breast cancer subtypes, namely basal-like, HER2 enriched, luminal A and luminal B, as defined in material and methods. As seen in Figure 1C, the basal-like subtype, followed by the HER2+ group, displayed the higher overexpression of genes within each signature. A reduced expression of genes included in the four signatures was clearly observed in luminal tumors, particularly in the luminal A subtype (Figure 1C). These data demonstrate that dysregulation of DDR pathways is more present in basal-like and HER2 positive breast tumors compared with luminal subtypes.

### 3.2. Analysis of Breast Cancer Subtypes Display Different Expression of Transcripts and Amplification of Genes

Analyses of raw transcriptomic data confirmed the upregulation of most of the genes involved in the gene set, especially in basal-like tumors and in the HER2-enriched subgroup (Figure 2A). Appendix A provides the full list of genes and their fold change (FC) compared with normal breast tissue; Appendix A displays the full list and the percentage of amplified genes. We identified a substantial number of genes that were amplified in breast cancer patients, being most of them included in the basal-like (69.5%) and HER2 (63.2%) breast cancer subtypes (Figure 2B). To get insights into those that were upregulated and amplified, we explored the correlation between both parameters in each specific breast cancer group (Figure 2C). To check for clinical relevance, we selected only those that were amplified in more than 5% of cases and overexpressed more than 1.5 times in tumors compared with normal breast. As shown in Figure 2C, only 19 genes met these criteria, and five of them were common to all breast cancer subtypes, including *NBN, PRKDC, RFWD2, UBE2T*, and *YWHAZ*. It is worth mentioning that, most of the highlighted genes above with more than 15% frequency of amplification appeared only in the basal-like and/or HER2+ breast cancer subtypes, including *YWHAZ, BRIP1, CCNE1, CDC6, DCLRE1C, NBN, RFWD2*, and *CSTF1*. A small number of amplified genes was observed in luminal tumors, particularly in the luminal A subtype, where the most frequently observed genes were *YWHAZ* and *NBN*, which reached 12% (Figure 2C).

### 3.3. Interacting DNA Damage Network Associates with Detrimental Prognosis

Taking in consideration that the identified gene sets share an elevated number of transcripts with overlapping biological functions, we explored the protein–protein interaction network of their components. The proteins coding by these genes showed a high protein-protein interaction coefficient (Cluster coefficient 0.7, PPI enrichment *p* value <1.0 × 10^−16^), confirming that most of them participate in redundant functions (Figure 3A).

To evaluate the role of the combination of these genes in relation to clinical outcome, we took advantage of the online tool KM Plotter, which associates gene expression levels with patient prognosis. The analysis of individual genes showed that high expression predicted detrimental relapse free survival (RFS) (Figure 3B) and overall survival (OS) (Figure 3D) in the whole breast cancer population, and in most of the breast cancer subtypes. Even considering that some individual genes did not predict for outcome (Figure 3B,D), we decided to use the 19 genes together as a signature, due to the high PPI observed and their significant expression in breast cancer (Appendix A), to explore their potential impact on clinical survival. The combination showed a stronger association with poor prognosis for RFS (HR: 1.64, CI 1.4–1.93, log rank *p* = 9.6 × 10^−10^) (Figure 3C), and OS (HR: 1.81, CI 1.32–2.48, log rank *p* = 0.00018) (Figure 3E) compared with each gene individually. We confirm these results using RNA seq data from TCGA in Appendix A for RFS (HR: 1.77, CI 1.13–2.77, log rank *p* = 0.012) and Appendix A for OS (HR: 1.47, CI 106–2.04, log rank *p* = 0.022). When evaluating the signature in different breast cancer subtypes, we observed that high expression of the 19-gene signature, observed in all molecular subtypes (Appendix A), correlated with detrimental RFS (luminal A, HR: 1.56, CI 1.22–1.99; log rank *p* = 0.00036; luminal B, HR: 1.43, CI 1.06–1.95, log rank *p* = 0.02; HER2+, HR: 1.71, CI 1.08–2.7, log rank *p* = 0.02), with more effect in the basal-like subgroup: basal, HR: 2.31, CI 1.65–3.22, log rank *p* = 4.2 × 10^−7^ (Figure 4A). A similar trend was observed for OS in all breast cancer subtypes (luminal A, HR: 1.74, CI 1.03–2.93; log rank *p* = 0.035; HER2+, HR: 2.57, CI 1.15–5.73, log rank *p* = 0.017; basal, HR: 2, CI 1.04–3.83, log rank *p* = 0.034 luminal B, HR: 3.45, CI 1.42–8.36, log rank *p* = 0.0036, (Figure 4B).

### 3.4. Amplified Genes Correlated with Poor Prognosis

Next, we decided to reduce the list of the identified genes by selecting only the 5 common genes, *NBN, PRKDC, RFWD2, UBE2T*, and *YWHAZ*, that met our initial criteria for clinical relevance (amplified in more than 5% of cases and with a transcriptomic FC expression of more than 1.5 in all breast cancer subtypes) (Appendix A). The higher expression of this 5-gene signature in breast cancer was significant (Appendix A). The combined analysis of these genes showed a strong association with poor RFS (HR:1.65 CI 1.41–1.93, log rank *p* = 1.6 × 10^−10^) (Appendix A) and OS (HR:2 CI 1.46–2.74, log rank *p* = 1.2 × 10^−5^) (Appendix A) in the whole breast cancer group. The worst outcome of patients with higher expression of these 5 genes was confirmed using a different data set in Appendix A for RFS (HR:1.77 CI 1.15–2.73, log rank *p* = 0.0091) and Appendix A for OS (HR:1.56 CI 1.13–2.15, log rank *p* = 0.0068). Appendix A describes the function of each gene.

For each specific breast cancer subtype, we observed that this signature was significantly overexpressed compared with normal tissue (Appendix A), and was associated with detrimental RFS in all subtypes (Figure 5A) (luminal A, HR: 1.57, CI 1.23–2.01; log rank *p* = 3 × 10^−4^; luminal B, HR:3.78, CI 1.64–8.69, log rank *p* = 0.00079; HER2+, HR: 3.2, CI 1.45–7.08, log rank *p* = 0.0024; basal, HR: 1.78, CI 1.27–2.49, log rank *p* = 0.00063). For OS, findings in the similar direction was observed (Figure 5B): luminal A, HR: 1.81, CI 1.09–3.02, log rank *p* = 0.021; luminal B, HR:3.78, CI 1.64–8.69, log rank *p* = 0.00079; HER2+, HR: 3.2, CI 1.45–7.08, log rank *p* = 0.0024; basal, HR: 2.18, CI 1.14–4.18, log rank *p* = 0.016.

Using a permutational analysis, we selected only those genes that stronger predicted clinical outcome in each subtype. For the basal-like subtype, *CSTF1, RAD1*, and *YWHAB* correlated with a detrimental OS (HR:3.61, CI 1.79–7.28, log rank *p* = 0.00013) and RFS (HR:1.68, CI 1.21–2.34, log rank *p* = 0.0019) (Figure 6A). In the case of HER2+, the overexpression of *FANCD2, MCM7, YWHAB*, and *BHLHE40,* provided an incremented risk of death (HR:3.92, CI 1.73–8.89, log rank *p* = 0.00042) and relapse (HR:2,11, CI 1.33–3.36, log rank *p* = 0.0012) (Figure 6B). Similar findings were observed in luminal B tumors with *BRIP1, CDC6, CSTF1, NBN, PRKDC, RFC4, RFWD2, UBE2T, YWHAB*, and *YWHAZ* (OS, HR:4.76, CI 1.96–11.56, log rank *p* = 0.00015; RFS, HR:1.64, CI 1.18–2.29, log rank *p* = 0.0032) (Figure 6C). Finally, the increase in expression of the genes *CCNE1, FANCI, RFC4, CDC6*, and *YWHAZ* associated with poor prognosis in luminal A breast cancer (OS, HR:2.54, CI 1.69–3.82, log rank *p* = 3.6 × 10^−6^; RFS, HR:1.84, CI 1.53–2.22, log rank *p* = 6.5 × 10^−11^) (Figure 6D).

## 4. Discussion

Identification of genomic dysregulated functions that predict prognosis and therefore can help to stratify patient risk to optimize therapeutic interventions, is a main goal in cancer research. In this context, genomic and transcriptomic programs differ among the different breast cancer subtypes and can constitute opportunities for biomarker identification, particularly when they are related to a specific biological function. This has been the case for the development of RNA-based panels that can stratify recurrence risk in early stage breast cancer patients [17].

In our work, transcriptomic studies revealed that functions involved in DNA damage such as those classified as *ATM, ATR, Fanconi Anemia*, and *BARD1* pathways are clearly dysregulated in breast tumors, particularly in the basal-like and HER2 subtype. The transcripts included within these gene sets were upregulated and correlated with those with gene amplification. These data showed that most of the amplified genes were included within the basal-like and HER2-enriched subtype. Some genes were commonly shared between subtypes like *NBN, PRKDC, RFWD2, UBE2T*, and *YWHAZ* (Appendix A) while others were specific of a subgroup, like *BLM, CCNE1, DCLRE1C, FANCD2, FANCI, MCM7, RAD1, RFC4*, and *RNF8* for the basal-like and *BHLHE40, BRIP1, CDC6*, and *CSTF1* for the HER2 positive subgroup.

The protein-protein interacting network showed a strong correlation between proteins coded by these genes, what demonstrates that although the gene sets were classified differently, most of the proteins have a common biological role and maintain some degree of redundancy. This is relevant as we aim to identify key regulators that could be susceptible for target inhibition. Moreover, the high clustering coefficient found in the PPI would render it more susceptible to drugs targeting its nodes. With this in mind, we evaluated the prognostic role of the nineteen genes included in this interacting network. The combined analysis showed a detrimental effect of their altered expression with respect to RFS and OS in the whole population and in different breast cancer subtypes, demonstrating the potential oncogenic role of the network. A relevant observation was related to the worse prognosis noted in those subtypes where less transcriptomic expression was observed, mainly in the luminal A subgroup. This can be explained as the presence of these genes is less frequent in this particular subtype, but once present they clearly denoted a tumorigenic behavior.

With the idea to translate these findings to the clinical setting, among the genes included in the PPI, we selected only those that were amplified in all subtypes. As demonstrated with the development of other prognostic biomarkers, RNA assessment using paraffin embedded tissue is a flexible and rapid manner to evaluate sets of genes [17]. The expression of these five selected genes correlated with detrimental prognosis in each breast cancer subtype. Those genes included *NBN, PRKDC, RFWD2, UBE2T*, and *YWHAZ*. While NBN is present in the four DNA damage pathways that are *ATR, Fanconi Anemia, ATM*, and *BARD1, YWHAZ* is just present in *ATR/ATM; PRKDC* in *BARD1*; and *UBE2T* in *Fanconi Anemia* pathway, respectively.

NBN is a component of the MRE11-RAD50-NBN (MRN complex) [26] which plays a critical role in the cellular response to DNA damage, and the maintenance of chromosome integrity, since it is involved in double strand break (DSB) repair, DNA recombination, maintenance of telomere integrity, cell cycle checkpoint control and meiosis [27]. The role of the MRN complex in tumorigenesis and cancer treatment has been already discussed [28]. PRKDC gene encodes for a 469 kDa protein called DNA-PK [29]. This protein forms part of the phosphatidylinositol 3-kinase-related family of protein kinases and is abundantly expressed in almost all mammalian cells [29]. It is involved in DNA non-homologous end joining required for DSB repair and V(D)J recombination [30,31]. Since DNA-PK is a critical component of the damage response machinery and taken in consideration that a high number of cancer treatments produce irreparable DNA damage, its expression correlates with decreased response to DNA damaging agents, and therefore therapeutic resistance in multiple cancers [32,33,34], including breast [35,36]. *RFWD2 and UBE2T* are two genes involved in ubiquitination. RFWD2 ubiquitination leads to subsequent proteasomal degradation of target proteins [37], being among them p53 [38] and c-Jun [39]. On the other hand, UBE2T is the E2 ubiquitin-conjugating enzyme for the *Fanconi Anemia* pathway [40] and monoubiquitinates several proteins of the pathway, that also appeared in our screening like FANCD2 and FANCI [41]. It is also involved in BRCA1 downregulation [42], one key protein with a role in breast cancer tumorigenesis. Finally, *YWHAZ* codes for an adaptor protein that belongs to the 14-3-3 proteins family [43]. It is implicated in the regulation of a large spectrum of signaling pathways including cell growth, cell cycle, apoptosis, migration, and invasion [44,45,46].

A relevant finding is that all the identified genes belonged to the same PPI network that includes genes with different molecular roles but that participate in the same biological function, in relation to orchestrate an adequate DNA damage response. In this context, some proteins participate in the ubiquitination process while others belong the kinase family of enzymes. This also suggest the redundancy of processes with the goal to maintain the integrity of the system. 

Identification of genomic correlates of a particular function is a main goal in cancer to better select patients for a given treatment. In the immunotherapy field, transcriptomic signatures to identify hot and pro-active immune tumors have been reported and some were able to predict response to check point inhibitors [47,48,49,50]. In a similar manner, some signatures have been described to predict response to PARP inhibitors and consensus panels have provided recommendations for assessment [51]. These studies aimed to identify responsive patients to a particular given treatment, but not to find tumors in which a specific biological function was more important and therefore could constitute a druggable vulnerability.

We recognize that future studies should be performed. Our group is currently exploring the role of these transcripts in relation to preclinical efficacy of these agents, and we are designing a study using human samples from patients treated with PARP inhibitors. 

## 5. Conclusions

In summary, we describe a DNA damage transcriptomic signature that discriminates patients with detrimental prognosis. Translational exploiting of these biomarkers to predict outcome, as well as their value to select more effective therapies, should be explored further.

## Figures and Tables

**Figure 1 cancers-13-02117-f001:**
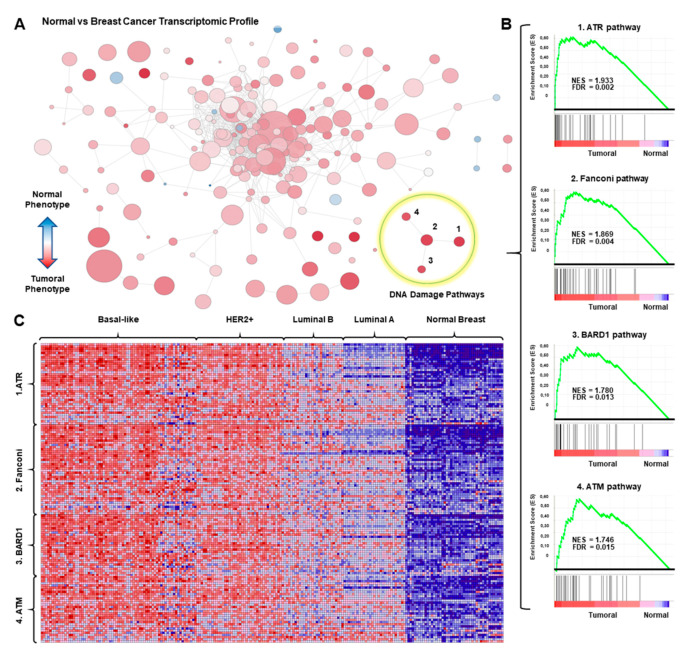
Upregulation of DNA damage pathways in breast cancer. (**A**) Gene set enrichment network comparing normal versus breast tumoral tissue. All gene sets from the Pathway Interaction Database are included, those overexpressed in the tumoral phenotype, are displayed in shades of red, and those overexpressed in the normal phenotype are displayed in shades of blue; while the size of the circle indicates the number of genes within the gene set. (**B**) ES Score profile and locations of DNA Damage pathways “*ATR* pathway”, “*Fanconi Anemia* pathway”, *“BARD1* pathway,” and “*ATM* pathway” members on the rank ordered list. Positive NES defines tumoral phenotype enrichment. (**C**) Blue–red diagram of all the genes that composed the DNA Damage pathways showing overexpressed genes in shades red, and downregulated genes in shades of blue.

**Figure 2 cancers-13-02117-f002:**
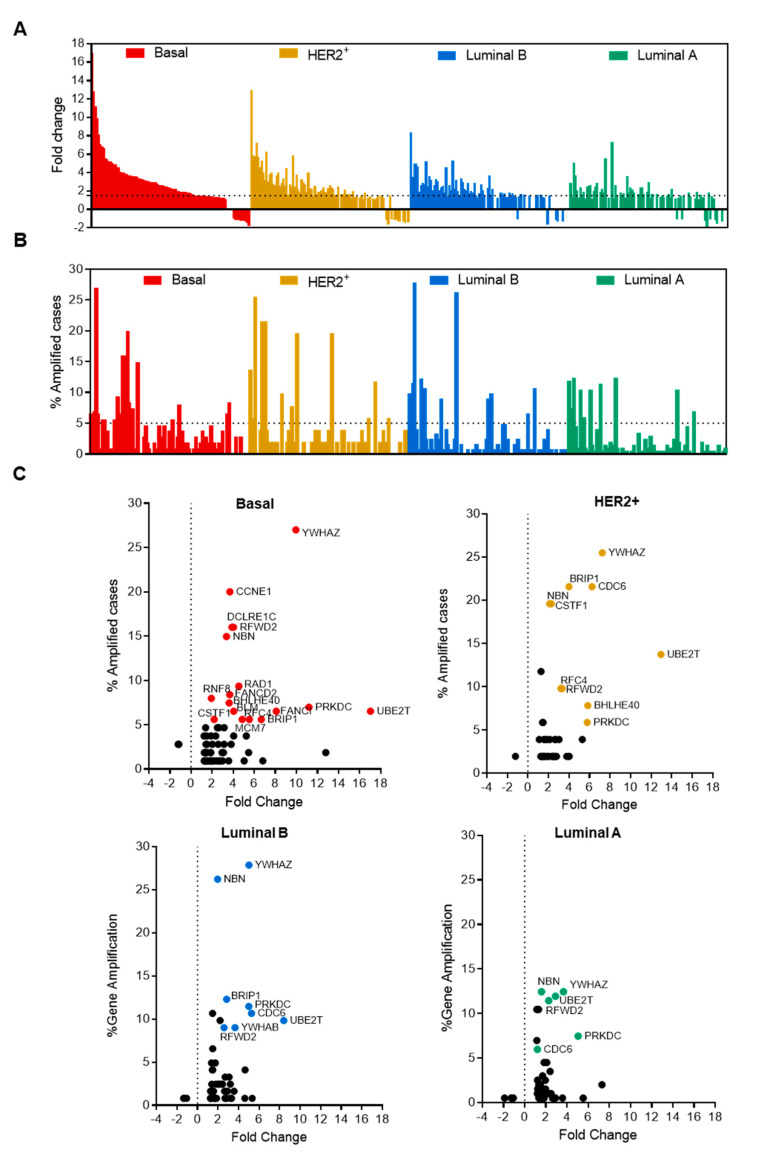
Amplification and expression change of DNA damage response genes in different breast cancer subtypes. (**A**) Bar graph showing fold change expression or (**B**) the percentage of amplified cases of genes involved in the *ATR, Fanconi Anemia, ATM*, and *BARD1* pathways in each breast cancer subtype (red, basal-like; yellow, HER2^+^; blue, luminal B and green, luminal A). The dot line represents the threshold we mark for each alteration. (**C**) Dot plot of DNA damage response genes showing fold change expression in X-axes and percentage of amplified cases in Y-axes for each molecular subtype. Those genes that exceed both thresholds are highlighted following the same color criteria than in A.

**Figure 3 cancers-13-02117-f003:**
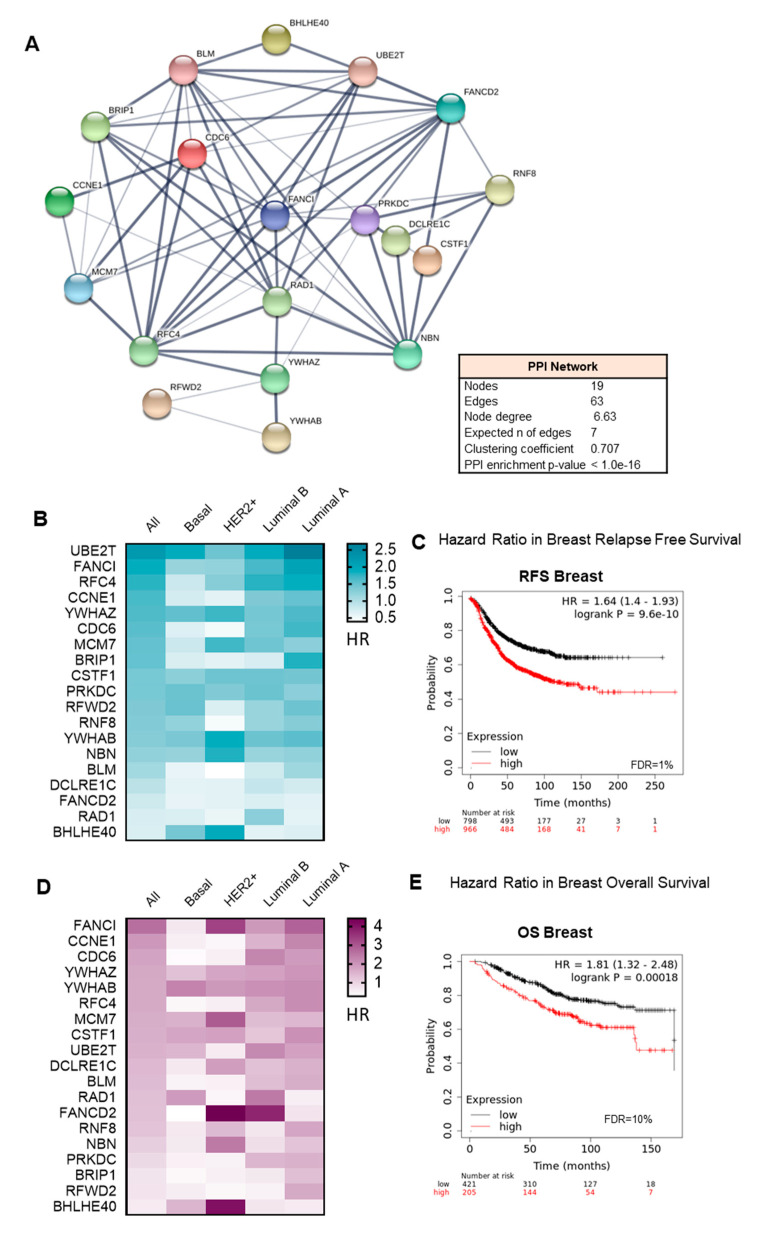
PPI map and functional annotation of bad prognosis-associated upregulated genes in breast cancer patients. (**A**) Protein–protein interaction map displaying the significant functional network integrated by the selected genes. (**B**) Heatmap showing the HR of the expression of individual genes for the relapse free survival (RFS) in breast cancer and its subtypes. (**C**) Association with RFS of combined genes in breast cancer. (**D**) Same as B, but for overall survival (OS). (**E**) Same as C for OS.

**Figure 4 cancers-13-02117-f004:**
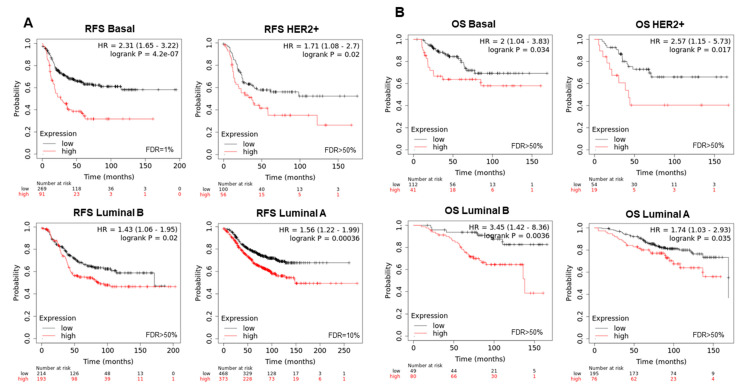
Association between gene expression and a worse survival across intrinsic subtypes of breast cancer. (**A**) Association with RFS and (**B**) OS with the expression of 19 selected genes (*BHLHE40, RFWD2, BRIP1, PRKDC, NBN, RNF8, FANCD2, RAD1, BLM, DCLRE1C, UBE2T, CSTF1, MCM7, RFC4, YWHAB, YWHAZ, CDC6, CCNE1*, and *FANCI*) included in the response to DNA damage in all breast cancer subtypes.

**Figure 5 cancers-13-02117-f005:**
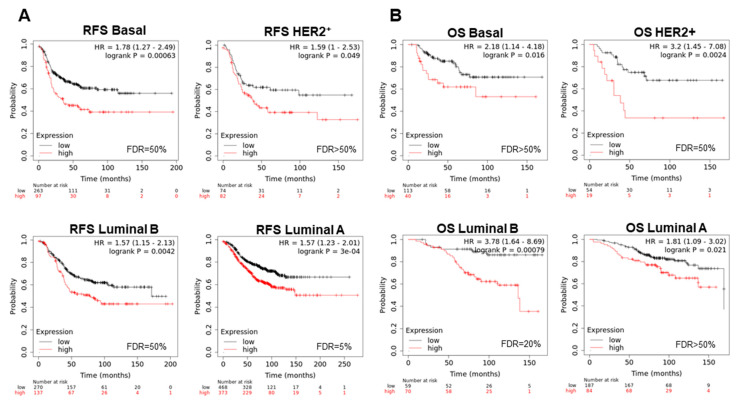
Association between expression of shared genes and a worse survival across intrinsic subtypes of breast cancer. (**A**) Survival analysis for 5 genes that are commonly altered in all breast cancer subtypes (*YWHAZ, UBE2T, RFWD2, PRKDC,* and *NBN*) with RFS (**B**) and OS across different subtypes.

**Figure 6 cancers-13-02117-f006:**
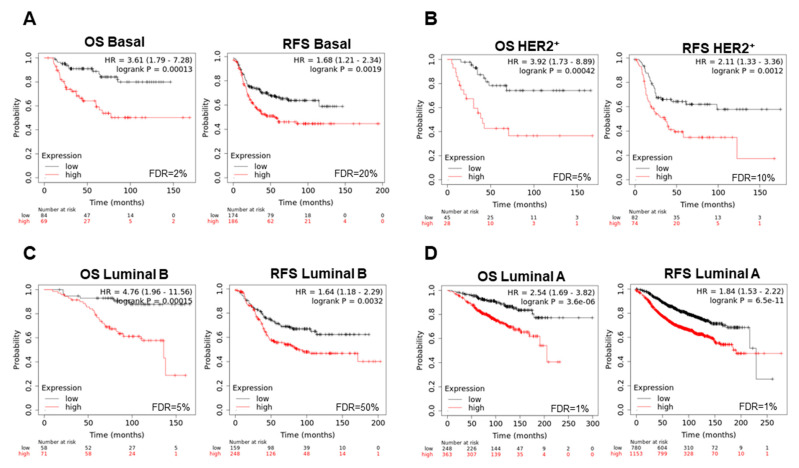
Best signature of DNA damage response genes to indicate poor survival in each breast cancer subtype. (**A**) Association with OS and RFS of gene sets in basal-like (*CSTF1, RAD1*, and *YWHAB*), (**B**) HER2+ (*FANCD2, MCM7, YWHAB,* and *BHLHE40*), (**C**) luminal B (*BRIP1, CDC6, CSTF1, NBN, PRKDC, RFC4, RFWD2, UBE2T, YWHAB*, and *YWHAZ*), and (**D**) luminal A (*CCNE1, FANCI, RFC4, CDC6*, and *YWHAZ*).

## Data Availability

The datasets, accessed in May 2018 and analyzed during the current study, are available in the following repositories: GSE21422 (https://www.ncbi.nlm.nih.gov/geo/query/acc.cgi?acc=GSE21422); GSE26910 (https://www.ncbi.nlm.nih.gov/geo/query/acc.cgi?acc=GSE26910); GSE3744 (https://www.ncbi.nlm.nih.gov/geo/query/acc.cgi?acc=GSE3744); GSE65194 (https://www.ncbi.nlm.nih.gov/geo/query/acc.cgi?acc=GSE65194); GSE42568 (https://www.ncbi.nlm.nih.gov/geo/query/acc.cgi?acc=GSE42568).

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
