# Peer review of "Genomic Correlates of DNA Damage in Breast Cancer Subtypes"

_cancers, 2021, doi:10.3390/cancers13092117_

Round 1

Reviewer 1 Report

A well presented paper showing the power of in silico analysis to identify the pathways involved in oncogenesis and predict clinical outcome. The results are well presented and clearly explained. I would have liked more detail about the genes involved in DNA damage and how the differentially expressed genes were mapped back to these pathways for those less familiar with in silico techniques.

Author Response

We appreciate the comment of the reviewer. We have made an effort to explain in more detail the role of the genes identified and how they interact together. In this context, we have included a paragraph in the discussion section (419-424) describing the importance of the protein-protein interacting (PPI) network of the identified genes in relation to the different molecular functions pf the identified genes. Although some genes belonged to ubiquitination or habour kinase-related activity, they all contribute to e to maintein the DNA integrity. 

Reviewer 2 Report

In this paper, authors explored the genomic landscape of breast cancer subtypes and identified a transcriptomic signature that selected 96 patients with a particular detrimental outcome. They also identified genes including NBN, PRKDC, RFWD2, UBE2T, and YWHAZ for clinical relevance. Following my comments and suggestions:

The work is well described and the design goal of the authors well developed. The authors thoroughly investigated the genomic landscape of breast cancer subtypes using various tools, including GSEA for transcriptome analysis and STRING for investigating the protein-level interactome. They also identified a number of genes and associated them with clinical data in a very significant way.

My major concerns regard above all the fact that the authors exclusively used a bioinformatics approach based on data present in the public database. 

In my opinion, the work would be much more relevant if the authors may couple some experimental data to the bioinformatics approach that could validate the data obtained.

For example, IHC (immuno-histochemical) experiments of selected genes (2 genes minimum) on a small cohort of breast cancer patients could show that the results obtained have experimental evidence at a clinical level. In alternative, the authors may validate the transcriptomic data by analysing experimentally, the quantitative expression profiles of selected genes in a cohort of breast cancer patients or in RNA isolated from breast cancer cell subtypes.

Author Response

We completely agree with the reviewer that confirmation of our results at a genomic and proteomic level in a cohort of breast cancer patients would undoubtedly increase the value of the manuscript. In this regard, we are currently executing that part, but taking in consideration the time that this process will take, we expect to have the results no shorter than in 6-12 months. This clearly limits the inclusion of that piece of data in this manuscript.

However, we have made an effort using other datasets to confirm the results obtained here. We have used an additional dataset and we have incorporated new confirmatory studies that have been included as a supplementary figure. For the whole signature, the RFS and OS data has been included as supplementary figure 2. For the set of five genes, this outcome analysis has been included as supplementary figure 3.

In addition, we have confirmed that the expression level of the selected genes were also higher in tumor samples compared with non-transformed breast tissue and it is shown inthese two supplementary figures. This information has also been included and explained in the results section (paragraphs: 249; 253-255; 256-257; 268-269; 272-275; 277-278). Furthermore, we have written a new paragraph (133-138) in material and methods section.

Round 2

Reviewer 2 Report

Dear authors, I red your comments and analyse your amends in the manuscripts. As written in the first revision, the work is well described and the design goal of the authors well developed. The authors also identified a number of genes and associated them with clinical data in a very significant way.

My major concerns regarding the methodology regarding the exclusive use of bioinformatics are still standing, but it is true that an experimental validation would require the retrieval of samples that probably are currently not available or that are the subject of other or further work.